



# The impact of synoptic storm likelihood on European subseasonal forecast uncertainty and their modulation by the stratosphere

Philip Rupp[1], Jonas Spaeth[1], Hilla Afargan-Gerstman[2,3], Dominik Büeler[2,4], Michael Sprenger[2], and Thomas Birner[1,5]

[1]Meteorological Institute Munich, Ludwig-Maximilians-University, Munich, Germany
[2]Institute for Atmospheric and Climate Science, ETH Zürich, Switzerland
[3]Faculty of Geosciences and Environment, Univeristy of Lausanne, Switzerland
[4]Center for Climate Systems Modeling (C2SM), ETH Zürich, Switzerland
[5]Deutsches Zentrum für Luft- und Raumfahrt (DLR), Institut für Physik der Atmosphäre, Oberpfaffenhofen, Germany

**Correspondence:** Philip Rupp (philip.rupp@lmu.de)

**Abstract.**

Weather forecasts at subseasonal-to-seasonal (S2S) timescales have little or no forecast skill in the troposphere: individual ensemble members are uncorrelated and span a range of atmospheric evolutions that are possible for the given set of external forcings. The uncertainty of such a probabilistic forecast is then determined by this range of possible evolutions – often quantified in terms of ensemble spread. Various dynamical processes can affect the ensemble spread within a given region, including extreme events simulated in individual members. For surface pressure or geopotential height forecasts over Europe, such extremes are mainly comprised of synoptic storms propagating along the North Atlantic storm track. We use ECMWF re-forecasts from the S2S database to investigate the connection between different storm characteristics and ensemble spread in more detail. We find that the presence of storms in individual ensemble members at S2S time scales contributes about 20% to the total geopotential height forecast uncertainty over Northern Europe. Furthermore, certain atmospheric conditions associated with substantial anomalies in the North Atlantic storm track show reduced geopotential height ensemble spread over Northern Europe. For example, during periods with a weak stratospheric polar vortex, the genesis frequency of Euro-Atlantic storms is reduced and their tracks move equatorwards. As a result, we find weaker storm magnitudes and lower storm counts, and hence anomalously low subseasonal ensemble spread, over Northern Europe.

## 1 Introduction

Weather prediction at subseasonal-to-seasonal (S2S) timescales remains a significant challenge in meteorology, particularly for forecasts of the extratropical troposphere (e.g. White et al., 2017). Forecast skill is highly limited at these time scales and ensemble forecasts typically aim to model the distribution of possibilities into which the real atmosphere can evolve for a given set of external forcings and boundary conditions. Here, boundary conditions and forcings can refer to, e.g., prescribed sea surface temperatures or radiation inputs, which set the outcoming distributions of possible dynamic evolutions of the atmosphere. However, boundary conditions of a sub-system, like the troposphere, can also be given by the time-evolving state





of a different system if they evolve on different time scales: e.g. the typically slowly evolving stratosphere may be thought of an upper boundary condition for the more quickly evolving troposphere. In that sense, initial conditions of certain sub-systems may effectively serve as boundary conditions to other sub-systems.

The ensemble spread of a probabilistic S2S forecast provides a measure of the predictability of the system in a specific situation, assuming that model errors are small. Forecasts with large spreads correspond to situations with a wide range of possible scenarios and are hence associated with high uncertainty about the evolution of the system, while forecasts with small spreads can be more certain about how the system will evolve. Various physical and dynamical processes associated with different spatial scales can act as sources of S2S variability (or uncertainty) in the Euro-Atlantic sector. On large scales, the

dominant mode of variability in that region is the North Atlantic Oscillation (NAO) (e.g. Hurrell et al., 2003; Benedict et al., 2004). Differences in NAO phase between ensemble members are associated with relative meridional shifts of the mid-latitude jet and its associated strong pressure gradients. The presence of such a strong gradient leads to a potential for large ensemble spread in mid-latitude forecasts, e.g., if different ensemble members predict the position of this gradient at different latitudes. On synoptic spatial scales, a major source of subseasonal variability is given by Rossby waves developing, propagating and

breaking on the mid-latitude jet. Dispersion in the phase or magnitude of these waves among different ensemble members can increase the ensemble spread of the S2S forecast. On more regional scales, the ensemble spread is strongly influenced by the development of extratropical cyclones (in this study simply referred to as storms). Such storms are typically generated and amplify over the baroclinic regions in the Western North Atlantic and travel eastward towards Europe, where they often decay. The aggregated paths of storms form the North Atlantic storm track. The present study aims to analyse the contribution of

storms to the spread of S2S ensemble forecasts over the Euro-Atlantic sector.

The above dynamical sources of subseasonal ensemble variability are generally coupled to each other. For example, the NAO is strongly coupled to the position and strength of the North Atlantic storm track and different NAO phases influence the development and evolution of North Atlantic storms. Generally, a positive NAO phase is linked to a poleward displacement of the storm track and a higher number of extreme cyclones (Pinto et al., 2009; Donat et al., 2010), while a negative NAO phase

is associated with an equatorward shift of the storm track and more frequent storm extremes in Southern Europe, especially over the Iberian Peninsula (Merino et al., 2016).

Certain sets of boundary conditions and external forcings can result in anomalous forecast spread, by modifying the likelihood or characteristics of the dynamical processes described above. Of particular interest are situations in which the ensemble forecasts converge toward a narrower range of possible evolutions (i.e., the forecast spread is unusually small), indicating a

period of enhanced predictability (so-called 'windows of forecast opportunity'). These windows are often linked to specific atmospheric configurations or phenomena that temporarily reduce variability within the forecast, such as dominant weather regimes, teleconnections, or ocean-atmosphere interactions. During these periods, the forecast skill is higher, offering valuable opportunities for planning and decision-making beyond the usual limits of S2S forecasting (e.g. Mariotti et al., 2020).

It has long been known that the state of the stratospheric polar vortex (SPV) can influence the dynamics of the troposphere

(Baldwin and Dunkerton, 2001). Due to the long characteristic time scales of the stratosphere, the SPV therefore acts as a source of predictability for surface weather in the Northern Hemisphere on S2S time scales (Baldwin et al., 2003; Domeisen



et al., 2020). The downward influence of SPV anomalies on the Euro-Atlantic sector involves an NAO signal (e.g. Blessing et al., 2005) and associated latitudinal shifts of the mid-latitude jet (Maycock et al., 2020) and North Atlantic storm track (Butler et al., 2017; Afargan-Gerstman and Domeisen, 2020; Afargan-Gerstman et al., 2024). The modification of the storm

track is further consistent with the feedback of tropospheric eddy behaviour to stratospheric polar vortex anomalies found by various authors (Hitchcock and Simpson, 2016; Domeisen et al., 2013; Rupp and Birner, 2021).

The described downward influence of SPV anomalies (negative NAO phase, southward shift of the North Atlantic storm track) manifests robustly as average over many cases. However, anomalies in SPV strength can also modify tropospheric variability and hence be associated with windows of forecast opportunity. Recently, Spaeth et al. (2024) have shown that

periods with weak SPV are followed by reduced S2S forecast uncertainty over Northern Europe. This reduction also translates into enhanced forecast skill (Domeisen et al., 2020; Büeler et al., 2020). Spaeth et al. (2024) suggest that this anomaly in forecast uncertainty results from a southward-shifted North Atlantic storm track and correspondingly reduced synoptic activity over Northern Europe. However, a weak SPV can potentially also modify other characteristics of the North Atlantic storm track, in addition to its latitudinal position (like the magnitude or the occurrence frequency of storms), which could further

contribute to the modulation of forecast uncertainty over Northern Europe.

Studies have further shown that blocked weather situations are often associated with a modified North Atlantic storm track (e.g. Vallis and Gerber, 2008; Yang et al., 2021), and hence might have an influence on Euro-Atlantic forecast uncertainty at S2S time scales. Spaeth et al. (Under revision) found anomalies in ensemble spread of subseasonal forecasts depending on the weather regimes dominant during initialisation. Regimes with a more blocked Atlantic jet were generally associated

with negative 1000 hPa geopotential height (Z1000) spread anomalies over Northern Europe and positive spread anomalies in surface temperature, while regimes with a more zonal Atlantic jet showed the opposite signals. They further suggested differences in synoptic storm activity as the predominant driver of these spread anomalies.

Given the potential of storms to contribute to subseasonal forecast spread, this study quantifies this connection in more detail. By systematically analysing the relationship between strong storm events and the variability in ensemble predictions,

we provide insights underlying mechanisms driving forecast uncertainty.

This manuscript is structured as follows: Section 2 describes the technical details of the dataset used and how we identify and track isolated storm features. Section 3 discusses an exemplary case of how the occurrence of an individual storm at S2S time scales can contribute to ensemble spread over Northern Europe, after which Section 4 analyses the general correlation between storm characteristics and ensemble spread. In Section 5 we quantify the influence of variations in the stratospheric

polar vortex on storm track characteristics and demonstrate how certain persistent teleconnection patterns can affect European ensemble spread. Finally, Section 6 discusses our findings and Section 7 summarises our main conclusions.



## 2 Model and data

### 2.1 Subseasonal forecasts

This study uses ensemble forecasts provided by the European Centre for Medium-Range Weather Forecasts (ECMWF) as part
of the S2S Predicion Project (Vitart et al., 2017). We use 10-member hindcasts (excluding the control member) corresponding
to 25 real-time forecasts initialised regularly throughout the winter period 1st December 2020 to 28th February 2021. Each
real-time initialisation provides hindcasts for the previous 20 years, which gives a total of 500 hindcast ensembles covering
the December-to-February (DJF) periods between 2000/01 and 2019/20. Within this manuscript, we analyse daily snapshots
of 1000 hPa geopotential height (Z1000) and mean sea level pressure (MSLP) fields provided on a $2.5° \times 2.5°$ regular grid.

The use of 10-member hindcasts allows us to highlight the impact of individual storms on the ensemble spread. However,
this process is equally important in larger ensembles, e.g., 51-member real-time forecasts. S2S forecasts aim to model the dis-
tribution of possible scenarios (including the actual evolution of the atmosphere), with extreme events like strong mid-latitude
storms forming the tails of this distribution. In undersampled forecasts, it is possible to obtain estimates of this distribution
in situations where storms are or are not predicted within the ensemble, hence allowing for a direct comparison of potential
alternative realities (with and without storms). In well-sampled forecasts, situations where no storm is predicted within an
ensemble are rare, but the general impact of the tail of the underlying probability distribution on the corresponding forecast
spread remains the same. Throughout this manuscript, we use ensemble spread in terms of ensemble variance to quantify
forecast uncertainty.

A daily climatology of our dataset is constructed as a leadtime-dependent average over all available forecasts, without any
additional smoothing. Anomaly fields for each member and for the ensemble spread are computed as deviations from this
climatology within the respective field.

### 2.2 Cyclone identification and tracking

We use a feature-based approach to identify extratropical cyclones (here simply denoted as storms) in the subseasonal model
runs. The algorithm, developed by Wernli and Schwierz (2006) and refined by Sprenger et al. (2017), detects closed contours
in mean sea level pressure (MSLP), enclosing one or several local MSLP minima. A time-dependent spatial storm mask is then
defined via the area enclosed by the outermost closed contour of a storm. We further define a corresponding storm centre as
the location of minimum MSLP within this closed contour and a storm strength as the value of this MSLP minimum ($P_{min}$).
Storm tracks (paths of the corresponding storm centres) are computed based on 6-hourly data, however, only daily values are
used to match the available Z1000 data. To neglect weak and short-lived storms, we only consider storms with a total lifetime
of at least 36 h and peak storm strength of $P_{min} < 985$ hPa along the track. For further information on the detection algorithm
see Sprenger et al. (2017). For an in-depth analysis of the biases in 6-hourly-based cyclone tracks in S2S forecasts see also
Büeler et al. (2024).





## 2.3 Stratospheric polar vortex states

The stratospheric polar vortex state associated with a forecast is defined based on the zonal mean zonal wind at 10 hPa and 60°
North ($U_{60}^{10}$ index) in the initial conditions. Forecasts with the lowest and highest 20% of initial $U_{60}^{10}$ are classified as weak and strong vortex forecasts, respectively. The remaining 60% of forecasts are classified as having a moderate vortex.

## 3 Case study of the connection between storms and forecast spread

Extratropical cyclones (or storms) and their associated fronts are devastating natural hazards and represent extremes of synoptic variability over the Euro-Atlantic sector. They can be associated with negative MSLP anomalies of several tens of hectopascals
and Z1000 anomalies of several hundred metres, especially during winter months. Hence, atmospheric conditions involving a strong active storm generally form extreme outliers and fall within the negative tail of the climatological Z1000 distribution over Europe.

At subseasonal-to-seasonal (S2S) timescales, tropospheric mid-latitude ensemble forecasts show very limited skill. However, the distribution of ensemble members at those leadtimes can, based on a perfect-model assumption, be interpreted as the
130 distribution of possible atmospheric states for the given set of boundary conditions and external forcings. In the mid-latitudes, extreme storms occur within the heavy negative tail of this distribution. Hence, when one or a few members in the ensemble simulate strong storms, these ensemble forecasts tend to be associated with a large ensemble spread, i.e., large uncertainty, in that region. Thus, the occurrence likelihood of storms generally couples to the forecast uncertainty.

To highlight the connection between the occurrence of strong storms and the associated uncertainty in subseasonal Z1000
forecasts over Europe, we consider in Figure 1, as an example, the Z1000 forecast initialised on 31st December 2007 and the evolution of the associated forecast spread in terms of ensemble variance. At early leadtimes ensemble spread (and hence uncertainty) of Z1000 increases rapidly until saturation after approximately two weeks, although some variability persists due to insufficient sampling within the 10-member ensemble. The small ensemble size allows us to further highlight the impact of storms forming within individual ensemble members, as can be seen on leadtime day 31. Here, a single ensemble member
(member 6) develops a strong storm over the target region in Northern Europe, leading to Z1000 anomalies of about -400m, while all other members show Z1000 anomalies between ±200m. This outlier produced by the presence of a storm leads to a sharp peak in ensemble spread, exceeding 25000m². Figure 1 further indicates which ensemble members model an active storm in the vicinity of the target domain at given leadtime-days (shown as red crosses). If we remove all those member-leadtime combinations associated with an active strong storm (i.e., essentially creating a dataset without storms), the ensemble variance
is significantly smaller on day 31. We further see an overall reduction of spread when removing storms from the forecast.

To understand the dynamical processes associated with this case study, Figure 2 shows the spatial distribution of Z1000 and ensemble spread anomalies in member 6 of the forecast at different leadtimes, as well as the centre position of strong storms at these leadtimes in all of the members. Member 6 simulates a strong storm (green, left-facing triangle) on day 29 at around 55°N and 25°W. The storm is characterised by a pronounced Z1000 minimum of roughly 400m magnitude during its peak
(also seen in Fig. 1). This feature in the single member results in large spread across all ensemble members: the vicinity of





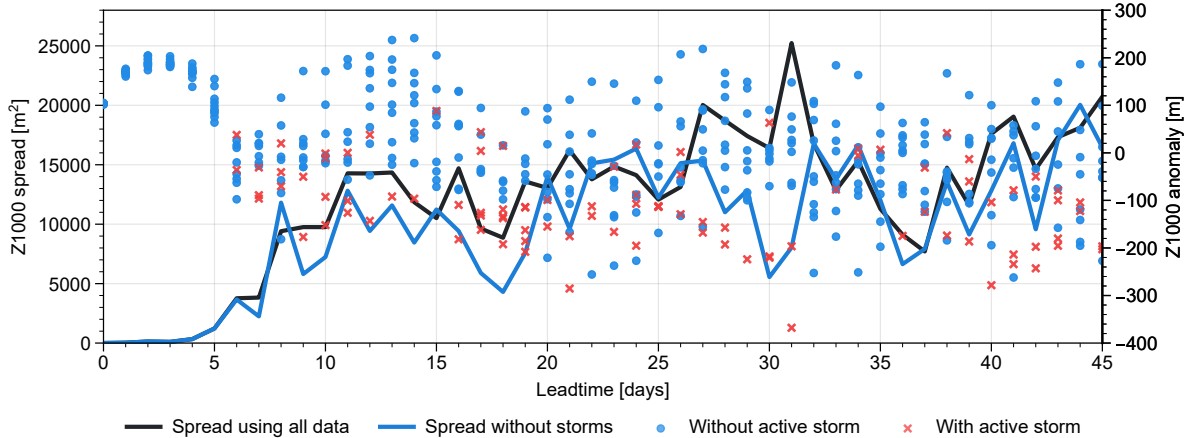

**Figure 1.** Evolution of Z1000 anomalies and associated ensemble spread averaged over Northern Europe (thick green box in Fig. 2) for the hindcast with initial conditions from 31st December 2007. Blue and red markers show Z1000 anomalies for each member and leadtime, with red crosses indicating member-leadtime combinations that exhibit a strong storm in the vicinity of the target region (thin green box in Fig. 2). Black line shows the ensemble spread of the full 10-member ensemble, blue line of the ensemble without accounting for storms (i.e. without red crosses). Note a strong storm with large negative Z1000 anomaly in member 6 on day 31.

the member-6 storm is characterized by a clear positive ensemble variance anomaly of about 25000m$^2$ magnitude. Over the next few days, the storm propagates eastward through our target region (purple box) over Northern Europe, with the spread signal closely following the storm. Figures 1 and 2 suggest that the ensemble variance would be substantially smaller if this one member did not develop a strong storm. On day 32, a new strong storm develops in a different member (blue right-facing triangle) around 55°N and 20°W, gradually strengthening over the next few days and again accompanied by a strong ensemble spread signature.

## 4   Systematic contribution of North Atlantic storms to forecast spread

The connection of strong mid-latitude storms with increased Z1000 ensemble spread over the Euro-Atlantic sector for a specific case was shown in Section 3. Here, we establish a more general connection between the occurrence of storms within an ensemble and anomalies in ensemble spread. To this end, Figure 3a shows a storm-centred composite averaged over all strong storms detected within the Euro-Atlantic sector. On average, strong storms (defined as in Section 3) have length scales of about 2000 km and magnitudes of around 200 m geopotential height anomaly. These large Z1000 anomalies lead to a heavy negative tail of the corresponding probability distribution. Within our 10-member ensembles, a single storm is therefore associated, on average, with an increase in ensemble spread of about 50% relative to the climatological spread (i.e. spread averaged over all available ensembles).



**Figure 2.** Z1000 anomalies (contours) and associated spread anomalies (shading) over the Euro-Atlantic sector during selected leadtime days in member 6 of the hindcast with initial conditions from 31st December 2007. Anomalies calculated as deviations from climatology (average over all ensembles, see Section 2). Markers indicate the centre location of strong storms in all of the members, with storms in different members shown as different marker shapes and colours. Marker sizes scaled with storm strength ($P_{min}$). Purple thick box indicates the Northern European region averaged over in Figure 1, green thin box shows the region used to identify members with active storm.





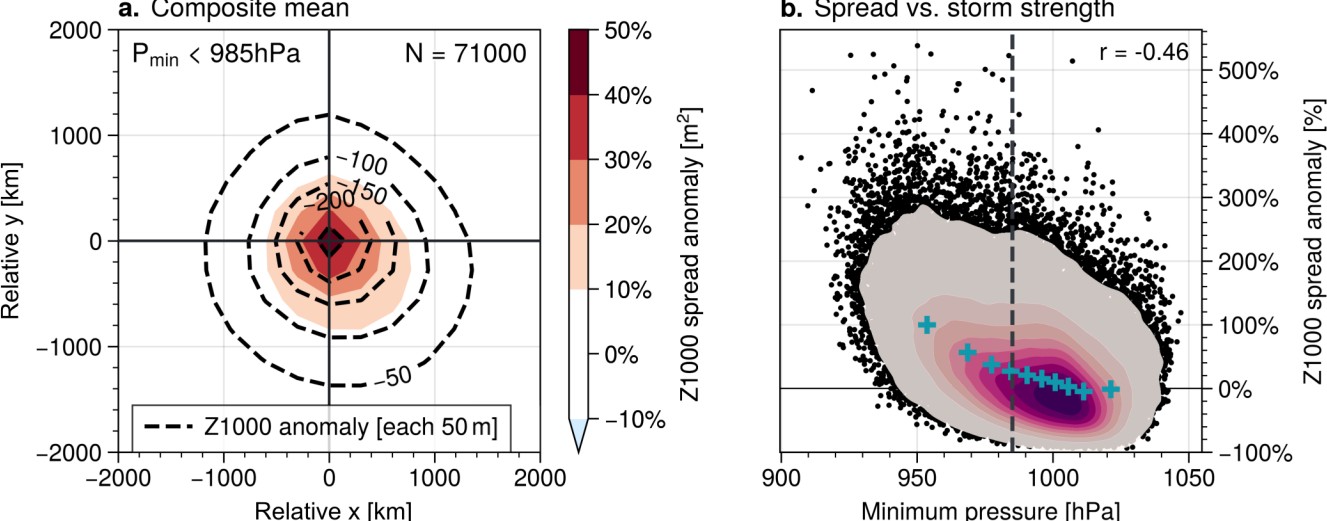

**Figure 3. a** Storm-centred composite, averaged over 71000 strong storms within the Euro-Atlantic sector during their day of maximum strength (i.e. minimum $P_{min}$). Composite is calculated in terms of zonal (x) and meridional (y) distance from the storm centre. Contours show Z1000 anomalies, shading shows spread anomaly relative to the climatological spread. **b** Scatter plot of Z1000 ensemble spread anomaly at the storm centre relative to climatology vs. corresponding storm strength as $P_{min}$. All storms in the Euro-Atlantic sector are considered. Black dots correspond to individual storms, shading visualises the distribution. Blue crosses show averages of subsets given by ten quantiles in $P_{min}$, i.e., along the x-axis. The correlation coefficient of $r = -0.46$ is based on all black points. Vertical dashed line indicates the threshold of 985 hPa used to classify strong storms.

While Figure 3a illustrates the average effect of individual storms on the ensemble spread, Figure 3b shows that the strength of this effect correlates with the strength of the storms (correlation coefficient $r = -0.46$). Stronger storms generally form larger negative tails of the Z1000 distribution and are therefore associated with increased spread anomaly. Figure 3b further shows that the effect is only present for strong storms, that form extreme values in Z1000.

The extreme Z1000 signature of storms intuitively is related to the spread within ensemble forecasts. However, it is not clear how strong this effect is and what its quantitative contribution is to the overall spread. Figure 4a shows a lag-composite of large-spread events over Northern Europe. While the increase in ensemble spread at lag day 0 is per construction, the associated Z1000 anomaly distribution is strongly skewed and associated with a heavy tail at extreme negative values. At the same time, Figure 4b shows the likelihood of storm occurrences over Northern Europe to be increased substantially and storms to be even

stronger (lower $P_{min}$) during these large-spread events. The signals in storm density and strength suggest increased storm activity to be a significant contributor to such events with extreme spread.

Figures 3a and 3b indicate a substantial contribution of synoptic storm activity to the formation of spread extremes. But how much impact do storms have on the overall mean spread within a subseasonal forecast? To answer this question, Figure 5 shows two versions of forecast spread evaluation over Northern Europe averaged over all available hindcasts: one version as



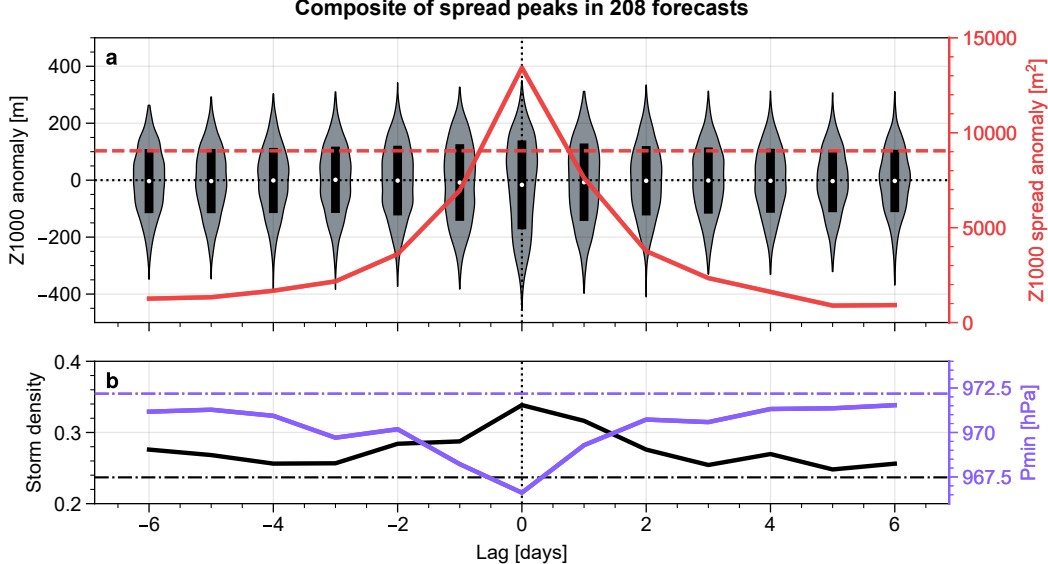

**Figure 4.** Lag-composite over 208 peaks in ensemble spread of Z1000 averaged over Northern Europe (purple box in Fig. 2). **a** distribution of Z1000 anomay as violin-plots and mean ensemble spread as solid red line. Horizontal dashed line indicates the climatological 95th percentile of spread. **b** composite mean evolution of storm density (in storms per member and day) and mean storm strength (as $P_{min}$) of storms (within green box in Fig. 2). Horizontal dash-dotted lines indicate climatological mean values.

a simple climatology including all data, and a version where all storms are disregarded when computing the spread. Storms are "removed" by not accounting for a certain data point if the corresponding member contains an active strong storm around our target region at the corresponding leadtime (as also shown in Fig. 1 for a single forecast). Without storms, the Northern European Z1000 ensemble spread drops by more than 20% at subseasonal leadtimes.

## 5    Influence of the stratospheric polar vortex on the North Atlantic storm track and Z1000 spread

We have established a systematic connection between storm activity and ensemble spread in Northern European Z1000 forecasts on S2S timescales, with storm activity contributing about 20% to the overall spread during winter. As a result, processes that modify the characteristics of the storm track (e.g. its position or shape) or individual storms (e.g. strength or occurrence frequency) should also project on the forecast uncertainty. In particular, this connection holds for large-scale teleconnections or climate change.

For example, Spaeth et al. (2024) found a systematic reduction in Z1000 ensemble spread over Northern Europe in S2S forecasts initialised during a weak phase of the stratospheric polar vortex. They further suggested that this reduction primarily results from a southward shift of the mid-latitude jet and associated North Atlantic storm track. Such an equatorward shift of the storm track is consistent with the negative phase of the NAO that is part of the canonical weak vortex signature (e.g.




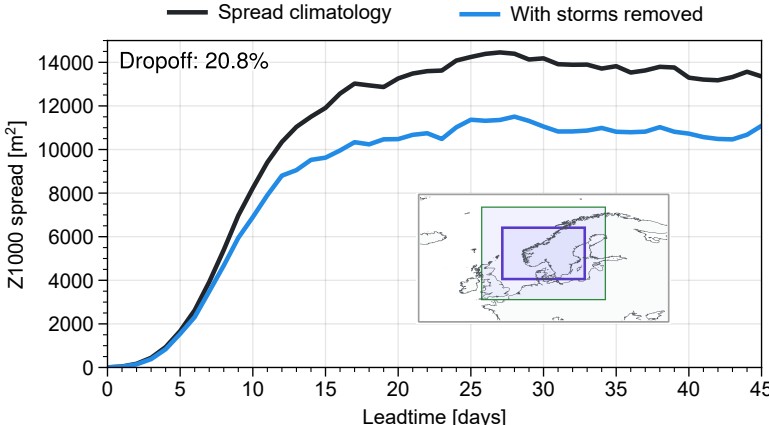

**Figure 5.** Evolution Z1000 ensemble spread averaged over Northern Europe (purple box) and over all forecasts. Shown is the spread computed based on all available data (climatology) and a dataset where member-leadtime combinations with an active strong storm over Northern Europe (green box) are removed. Removing storms leads to a significant dropoff (about 20%) of the spread during leadtime days 20-40.

Afargan-Gerstman et al., 2024). In this section, we investigate in more detail if and how changes in storm tracks and the

195 likelihood distributions of individual storms can contribute to the negative spread anomaly found over Europe following weak stratospheric polar vortex periods.

Figure 6a illustrates the changes in Northern European Z1000 ensemble spread depending on the stratospheric polar vortex state in S2S forecasts during initialisation. Weak polar vortex states are associated with substantially lower spread compared to strong polar vortex states. At the same time, Figures 6b and c show that fewer and weaker storms reach Northern Europe

following periods with weak polar vortex, while a strong polar vortex leads to more and stronger storms. As we have shown in Section 4, both storm density and strength can affect the Z1000 ensemble spread. All three quantities (Z1000 spread, storm density and storm strength over Northern Europe) show significantly larger differences between initialisations with weak and neutral polar vortex than between initialisations with neutral and strong vortex. This asymmetry could point to a fundamental difference in the downward influence of positive and negative polar vortex anomalies.

The reduction in storm density over Northern Europe during weak polar vortex periods (Figure 6b) arises in part due to a reduction in the total frequency of storm genesis over the Euro-Atlantic sector (Figure 7). In fact, the average number of strong storms developing per member and day over the Atlantic differs between weak and strong polar vortex initialisations with 17.0 and 19.8 storms, respectively (Fig. 7a). Furthermore, Euro-Atlantic storms are generally weaker (Fig. 7b) and slightly reduced in zonal velocity (Figure 7d) after periods with weak polar vortex.

To obtain more insights into how the stratosphere can influence the North Atlantic storm tracks, Figure 8a visualises the overall density of storms in forecasts at subseasonal leadtimes. The storm track clearly extends across the whole North Atlantic. Figures 8b and 8c further show the location of storm genesis and lysis (i.e., the regions where storm tracks start or end). Storms tend to be generated along the US east coast and dissipate on their way to Northern Europe. A hotspot of both genesis and



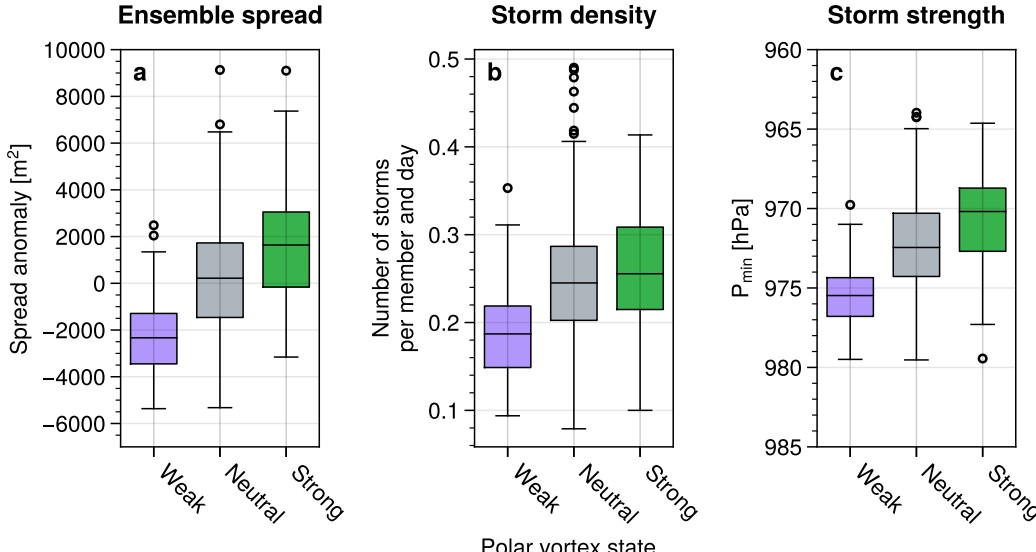

**Figure 6.** Boxplots showing the mean Z1000 ensemble spread anomaly (**a**), storm density (**b**) and storm strength (**c**) over Northern Europe (green box in Fig. 2) for hindcasts with varying stratospheric polar vortex strength. Hindcasts are classified as having a weak or strong polar vortex if their $U_{60}^{10}$ index at initialisation lies within the lowest or highest 20%, respectively, and as having a neutral vortex otherwise. Note the orientation of the y-axis in **c**.

lysis is visible near Southeast Greenland, likely due to interactions of the flow with prominent geographical features in that
area (e.g. Schwierz and Davies, 2003; Skeie et al., 2006). Figure 8d shows the change in storm activity due to weak SPV
conditions. Following weak vortex conditions, a clear anomalous dipole structure is visible, indicating a southward shift of the
climatological storm track. Figure 8e and f further support a similar pattern, with a more southward genesis and lysis of the
midlatitude storm track.

## 6 Discussion

This study analyses the impact of mid-latitude storms on S2S forecast spread in the Euro-Atlantic sector. As discussed in Section 2, we use 10-member ensembles to highlight the influence of individual storms on the forecast uncertainty. The influence of a single storm (such as in Fig. 3a) is generally dependent on ensemble size, as larger ensembles have a smaller chance of forecasting no or only a few storms on a certain day. However, the total influence of all storms within the system (as in Fig. 5) should be insensitive to the ensemble size, as it primarily depends on the shape of the underlying probability distribution for
the given set of boundary conditions and external forcings. To test this assumption, we performed a sensitivity analysis where we only considered 5 of the 10 members in each ensemble (not shown), and found the results in Figure 5 to be robust.



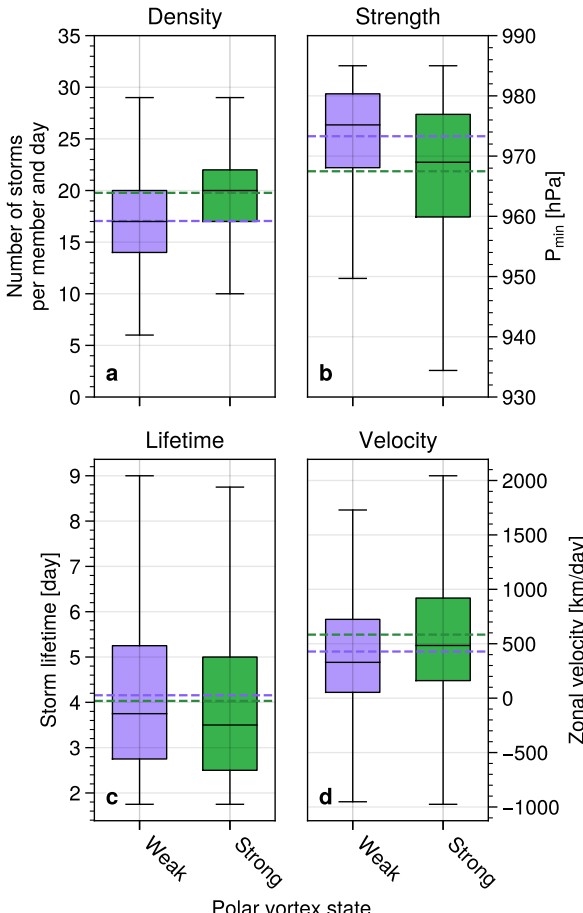

**Figure 7.** Different characteristics of strong storms within the Euro-Atlantic sector (80°W-40°E and 30°-80°N) within ensembles initialised with a weak or strong stratospheric polar vortex, defined as in Fig. 6. The distributions of storm density, strength, lifetime and velocity are shown. Storm strength is computed by the minimum pressure along the track, lifetime by the total number of days the storm was identified, and velocity via a linear fit of the distance in zonal direction the storm travelled within each six-hour timestep. All quantities are given for leadtimes 20 to 40 and averaged over the total number of members within each group. Horizontal dashed lines show the means of the corresponding distributions.



**Figure 8.** Density of storm centre locations during leadtime days 20 to 40. **a** density of the full tracks (all time steps combined) with black line indicating the mean track (mean latitude for each longitude bin), **b** density of genesis location (first step of each track) and **c** shows lysis location (last step of each track). Densities are given as fractions of storms within $7.5°\times7.5°$ bin compared to the total number of storms. Panels **d**, **e** and **f** differences between densities computed based on all ensembles (climatology) and ensembles initialised with a weak stratospheric polar vortex. Green line in d indicates the mean track during weak polar vortex conditions.





We demonstrate in Section 5 how the state of the stratospheric polar vortex influences the characteristics of the North Atlantic storm track. In particular, we show that a southward shift of the storm track following weak polar vortex conditions, together with a generally weakened genesis of storms over the North Atlantic, tends to lead to fewer and weaker storms (Fig. 7). These results are consistent with Afargan-Gerstman et al. (2024), showing that cyclones following strong SPV events tend to reach higher maximum intensities (i.e., lower sea level pressures) than the cyclones following sudden stratospheric warming (SSW) events.

Besides changes in strength and frequency, we do not find the storm lifetime to depend on different stratospheric states. However, there is a reduction of the average zonal velocity of storms by about 25% from 580 km/day following strong vortex conditions to 430 km/day following weak vortex conditions (Fig. 7d). This could indicate a general shortening of the storm track during weak vortex periods, although Figure 8c does not clearly indicate a shortening in terms of more westward cyclolysis. The weakening, southward shift and potential shortening of the North Atlantic storm track are all consistent with a reduction of storm count and strength over Northern Europe shown in Figures 6b and c. Since the results of Section 4 suggest a clear connection of storm activity to ensemble spread, the identified changes in storm track characteristics likely contribute to the anomaly in forecast spread over Northern Europe shown in Figure 6a and reported by Spaeth et al. (2024).

Furthermore, the troposphere has been shown to play a dominant role in the successful prediction of the downward influence of the stratospheric polar vortex, as e.g. indicated by Domeisen et al. (2020); Afargan-Gerstman et al. (2024). In this study we demonstrate that a representation of tropospheric circulation anomalies, in particular the response of mid-latitude storms, has a direct importance for reducing the uncertainty of S2S forecasts.

In mid-latitudes, other sources of S2S predictability originate from the coupling between tropics and extratropics. Modes of variability such as El Niño–Southern Oscillation (ENSO; e.g. Brönnimann, 2007; Zheng et al., 2019; Huang et al., 1998; Moron and Plaut, 2003), the Madden Julian Oscillation (MJO; e.g. Cassou, 2008) or the quasi-biennial oscillation (QBO; e.g. Wang et al., 2018) can persistently modify the mid-latitude circulation. If these changes in circulation affect the storm track, they are likely to further be associated with changes in forecast uncertainty. Furthermore, anthropogenic climate change has been shown to increase the likelihood for and magnitude of strong storms over Northern Europe (e.g. Knippertz et al., 2000; Pinto et al., 2007; Priestley and Catto, 2022) and, by extension, might hence lead to increased forecast uncertainty over Europe at S2S timescales.

In addition to extratropical storms, different representations of other dynamical features across the ensemble can lead to forecast uncertainty. For instance, the magnitudes and phases of synoptic-scale Rossby waves propagating on the mid-latitude jet will be strongly uncorrelated amongst different ensemble members at S2S timescales. The superposition of troughs and ridges of waves within different ensemble members at a fixed location contributes to the overall variability of the system and acts as a source of Z1000 ensemble spread. However, Z1000 anomalies associated with propagating Rossby waves are usually less extreme than those associated with strong storms. The relatively weak magnitudes, together with the large spatial scales, make synoptic Rossby waves harder to detect than storm features and to isolate their effect on ensemble spread.

The analyses within this study mostly assume a good representation of the North Atlantic storm track within the considered model, in which case signals in ensemble spread would be associated with proportional signals in forecast errors. If the model,



however, does not represent the storm track accurately, the spread-error proportionality does not hold and spread anomalies would partially correspond to an over- or under-confidence of the model. For example, Büeler et al. (2024) showed that the ECMWF model (see Sec. 2) has relatively small biases in cyclone frequency during boreal winter (the period analysed in the present study) at 3-4 weeks leadtime, but shows substantial biases over the Atlantic during the summer months. Our results suggest that such misrepresentations and model biases in terms of intensity or position of the storm track could be associated with anomalies in ensemble spread and forecast errors at S2S leadtimes.

## 7 Summary and conclusions

This study systematically quantifies the influence of synoptic storm event characteristics on subseasonal-to-seasonal (S2S) forecast spread over Europe. We find a significant contribution of storms to forecast uncertainty, with strong storms contributing about 20% of the mean Z1000 spread over Northern Europe. The analysis reveals that both the occurrence frequency and the strength of mid-latitude storms are crucial in increasing ensemble spread since stronger storms are associated with a heavier negative tail of the Z1000 probability distribution. Additionally, we find that certain atmospheric conditions, such as weak stratospheric polar vortex states, lead to decreased storm activity over Northern Europe, and are thereby associated with reduced ensemble spread in that region. The decreased storm activity is mostly due to a southward shift of the North Atlantic storm track, as well as a general decrease in storm strength and sparser genesis of storms over the North Atlantic. This suggests that the state of the stratospheric polar vortex significantly influences the predictability of weather patterns over Northern Europe and can open "windows of opportunity", where forecast uncertainty is substantially reduced. These results are consistent with the findings of Spaeth et al. (2024).

Isolating the role of extratropical storms for forecast uncertainty can improve storm forecasts and help to mitigate the risk due to extreme events on S2S timescales.

*Acknowledgements.* This research has been supported by the German Research Foundation (DFG) under grant no. SFB/TRR 165 (Waves to Weather). This project has received funding from the European Research Council (ERC) under the European Union's Horizon 2020 research and innovation programme (grant agreement No. 847456). HAG acknowledges funding from the European Union's Horizon 2020 research and innovation programme under the Marie Sklodowska-Curie (grant agreement No. 891514).

*Author contributions.* PR performed the main analyses and wrote the first version of this manuscript. JS contributed to the analyses and the interpretation of results. HAG and TB assisted with conceptualisation and interpretation. DB and MS computed the storm tracks and provided assistance with their interpretation. All co-authors contributed to the revision of the original manuscript.



*Data availability.* The forecasts used in this study are available via the S2S database hosted by ECMWF under https://apps.ecmwf.int/datasets/data/s2s.

Cyclone frequency datasets and other diagnostic codes are available from the corresponding authors upon request.

*Competing interests.* The authors declare that they have no conflict of interest. Some authors are members of the editorial board of Weather and Climate Dynamics.



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
