# Peer review of "The impact of synoptic storm likelihood on European subseasonal forecast uncertainty and their modulation by the stratosphere"

_EGUsphere, 2024_

## Author Comment (AC1)

We thank referee #1 for their review and various suggestions to improve the manuscript. In the following we will respond in to the different comments and explain the general changes we intend to make to the manuscript based on them. The reviewer's comments are in black italics, our responses are shown in blue. All line numbers and references refer to the originally submitted manuscript.

**GENERAL COMMENT**

*This manuscript analyses the relation between strong storms and forecast uncertainty, demonstrating how the prediction of storms, associated with strong geopotential height anomalies, affects the spread of ensemble forecasts in the Euro-Atlantic sector. Even though the ECMWF hindcast ensemble is small, as argued by the authors the results should naturally extend to larger ensembles.*

*This outcome is applied to the case of weak and strong stratospheric vortex state. The weaker frequency and depth of strong storms in the Euro-Atlantic sector (and Northern Europe) during weak stratospheric vortex conditions likely explains the decrease in uncertainty that follows these events.*

*Although the topic is original and interesting, the manuscript still requires substantial improvements in terms of linearity of the discussion and, ultimately, readability. Therefore, I encourage the authors to work specially on the structure of the text and on the precision of the langage, specially in the Introduction and Discussion sections. Other methodological and technical comments are included below. Once these issues have been addressed, I am willing to recommend publication of this work in Weather and Climate Dynamics.*

Thanks for your constructive review. We will revise the text based on your comments and try to improve readability and precision, especially for the introduction and discussion.

**SPECIFIC COMMENTS**

*Introduction and Discussion :*
*From reading these sections, I was expecting the whole work to be focused on SPV impacts on uncertainty. The main results, however, are those relating the presence of strong storms and forecast uncertainty. Could you work on these in order to make the focus of your work clearer to the reader ?*

Thank you for this valuable feedback. We understand the need for a clearer focus throughout the manuscript. In response, we will revise the Introduction and Discussion sections to better align with the main results of our study, which are centred around the general impact of strong storms on forecast uncertainty.

However, a significant area of applicability of our work lies in understanding how processes and teleconnections, such as the downward influence of the stratosphere, modulate forecast uncertainty through changes in the storm track. Originally, our study was strongly motivated by the anomalous spread signals following weak SPV states identified by Spaeth et al. (2024), but our results provide a more general quantification of the contribution of strong storms to forecast uncertainty.

*Both are rather dishomogeneous and difficult to follow. Connections between the different paragraphs and topics are often lacking.*

We will revise the text in these sections to improve the flow and better guide the reader through our study.

*Figure 1 : for an equal comparison between the ensemble spread with and without storms the size of the ensemble must be consistent. For example, for the "with storms" case you can compute a line (or better, a distribution of lines) where the ensemble includes storm cases, plus a number of random members necessary to equal the size of the "without storm" ensemble.*

*Figure 5 : see comment to Figure 1.*

We agree with the referee that adequate statistical treatment of ensemble forecasts are crucial to obtain robust scientific results. However, the main requirement towards ensemble sizes is a sufficient sampling to be able to draw robust statistical conclusions. Given large enough samples, the ensemble sizes do not necessarily have to be equal to make a direct comparison of statistical characteristics. In our case the sample sizes are indeed small. However, by choosing the variance we are using an unbiased-estimate of the ensemble spread, which leads to a robust quantification irrespective of the ensemble size. Hence re-sampling the distribution by adding "random members" drawn from that distribution will (on average) not change the computed variance.

We tried to illustrate this in Figure R1, which shows plots similar to Fig. 1 and Fig. 5 in the manuscript. Figure R1a shows the evolution of ensemble spread for the case study hindcast discussed in Section 3. The black line shows the Z1000 ensemble spread over Northern Europe for the full ensemble (all 10 members). The blue line shows the spread for the same dataset, but "without storms", so all members disregarded that exhibit an active storm at a given leadtime. The spread shown by the blue line is hence computed for a dataset with usually fewer than 10 members. Fig. R1a also shows as green dashed line a version of this "spread without storms", where we artificially increased the ensemble size to 10 members again (by randomly re-drawing members from the "without storm" distribution). It can be seen that the blue and the green dashed lines are almost identical, except for very small deviations at certain leadtimes. Since Fig. 1 in the manuscript is mainly used to visualise the impact of storms on the spread for an exemplary case, we do not believe that these small deviations do not have a substantial impact on the conclusions that can be drawn from the figure or the associated confidence of the reader.

In addition, the deviations between the blue and the green dashed line are not systematic, i.e., they have both positive and negative sign. Averages over many cases will therefore not show any difference between the different ensemble sizes. This is illustrated in Fig. R1b, where we computed the climatological evolution of the ensemble spread as function of lead time (similar to the black line in Fig. 5 in the manuscript), but for re-sampled ensemble sizes (by randomly drawing a certain number of members from the initially 10-member ensemble). It can be seen that on average, the evolution of the spread is essentially independent of ensemble size, up to a small increase in variability (of the variance) for very small ensembles. As argued above, this further suggests that the reduced ensemble size by removing certain members from the dataset does not impact the estimation of ensemble spread. To keep the methodology as simple as possible, we will therefore keep the approach used in the original study.

[Figure]

**Figure R1:** a) Same as Fig. 1 in the manuscript, but with an added line (green dashed) showing the ensemble spread for a dataset "without storm", but resampled to contain 10 ensemble members at each leadtime. The resampling is done by randomly drawing extra members from the set of the set of members "without storm" . b) Climatological evolution of ensemble spread (as shown in black in Fig. 5 of the manuscript) for different ensembles with reduced ensemble size (by randomly removing members from the dataset).

*Figures 6a and 7b : The fact that storm strength is high for small Pmin is confusing. Please address this issue as you think best.*

We will change the orientation of the y-axis in Fig. 7b, so it more reflects a 'storm strength', and add a corresponding note to the caption.

***TECHNICAL CORRECTIONS***

*Line 19 : "prescribed" or "initialised" in the case of S2S ?*

We    will modify the sentence accordingly.

*Lines 21-23 : Sentence is clear but quite convoluted. Please improve.*

We will split the sentence.

*Lines 32-33 : Do you want to say that uncertainties in the position of the jet (and its associated baroclinic zone) lead to a large ensemble spread in the mid latitudes ? Is there a difference in laying the focus on*

*the jet or on the strong pressure gradients ? In my opinion, they just correspond to different aspects of the same weather feature.*

We agree with the referee that the jet and the strong meridional pressure gradient over the Atlantic are aspects of the same atmospheric feature. The reason we introduced the pressure gradient is, that it might be easier for the reader to understand how differences in the location of this gradient across different ensemble members can lead to an increase in ensemble spread. We will revise the sentence to hopefully clarify the confusion.

*Line 34 : Maybe "synoptic-to-planetary scale" ?*

We will adapt this formulation.

*Line 36 : Since you are introducing the focus of this work, maybe make new paragraph ?*

We will add a line break.

*Line 57 : "NAO-like".*

Will be revised.

*Line 58 : "and OF THE North Atlantic storm track".*

Will be corrected.

*Line 59-61 : This sentence is rather vague. Please consider removing if outside of the scope and topic of this paper.*

We intend to remove the sentence

*Lines 90-93 : I find these two sentences confusing.*

We will revise the paragraph to clarify the details regarding ECMWF real-time forecasts and hindcasts.

*Why do you separate the 2020-21 hindcast from the hindcasts in previous years ?*

In our study we only utilise 10-member hindcasts, and none of the 50-member real-time forecasts. A practical reason is that we do not have storm track data available for the real-time runs. Another reason is, as explained in Section 2.1, that the impact of individual storms on the ensemble spread is easier to analyse in small ensembles. We will make sure the revised paragraph at the beginning of Section 2.1 clarifies the distinction between real-time forecasts and hindcasts.

*Could you clarify the sentence "Each real-time initialisation provides hindcasts for the previous 20 years". Is this again 10-member hindcasts over the previous 20 years ?*

Yes, for each real-time forecast ECMWF runs 20 hindcasts initialised at the same day-of-year as the real-time run, but in one of the preceding 20 years. These hindcasts have 10 members (plus a control member, which we do not use). More information on the ECMWF subseasonal forecast system can be found here: https://confluence.ecmwf.int/display/S2S/A+brief+description+of+reforecasts

*It would be useful to state explicitly which terminology is used to distinguish the runs initialised in 2000-2019 and the ensemble runs from 2020-2021, if there is a distinction. Also, what does the term "forecasts" in the rest of the text refer to ? Note the inconsistency with the term "hindcasts" used to present the dataset.*

We will make sure to clarify the introduction of "real-time forecasts" and "hindcasts", as they are typically used by ECMWF, and ensure these terms are used consistently within the manuscript.

*Line 123-124 : This sentence is an overstatement in my opinion. Maybe mention "intense" or "strong" storms ?*

Will be revised.

*Line 131 : Specify "Z1000 distribution".*

Will be revised.

*Line 135 : Specify the target domain from the beginning. Please add a motivation for choosing this region.*

Our target area is Northern Europe (55-67.5N and 0-20E), motivated by the region of reduced ensemble spread following weak stratospheric polar vortex events found by Spaeth et al. (2024). We will add this information to the text.

Spaeth, J., Rupp, P., Garny, H. and Birner, T., 2024. Stratospheric impact on subseasonal forecast uncertainty in the Northern extratropics. *Communications Earth & Environment*

*Line 143 : How is the vicinity of the storm to the target domain defined ?*

We define the vicinity of the target domain as within 5 degrees around the border. We will add this information.

*Line 158 : The case study is for "Northern Europe"   right?*

Yes. We will revise the statement.

*Line 161 : You have not yet defined the "Euro-Atlantic sector".*

We will add this information.

*Line 163 : Specify "a single storm CENTRE is therefore...". Further away from the centre, the percentage increase in spread is weaker.*

Will be revised.

*Line 172 : Could you justify why you go back to analysing Northern Europe, since the previous analyses in this subsection referred to the Euro-Atlantic sector ?*

As explained in the response for another comment, the main motivation for studying Northern Europe is the spread signal following weak stratospheric polar vortex states found by Spaeth et al. (2024), which was suggested to result from changes in storm activity. Figure 3 analyses the effect of strong storms on

the ensemble spread for storms detected within the entire Euro-Atlantic sector to show a more universal connection, which is not necessarily confined to Northern Europe.

*Lines 238-240 : A bit out of the blue here. Specify connection with previous or shift to a different paragraph.*

We will reformulate the sentence to better embed it into the paragraph.

*Line 244-246 : Acknowledge briefly that reduction in uncertainty does not necessarily mean an improvement in the predictions.*

We will add a brief statement regarding the improvement of predictions.

*Line 257-258 : It's a bit strange to consider Rossby waves and storms in opposition, since storms can be seen as Rossby-wave troughs. Please consider this while revising the text.*

We agree that synoptic Rossby waves and storms are highly correlated and, in particular, storms often develop in the troughs of Rossby waves during their non-linear breaking phase. In this study, we distinguish conceptually between the quasi-linear propagation stage of Rossby waves and storms since the two flow features affect forecast uncertainty in slightly different ways. Storms are usually isolated features, which at subseasonal leadtimes can occur in one ensemble, while another one does not show any signs of storms. We will add a note to the paragraph to clarify this matter.

*Line 259 : Correct the syntax of the last part of the sentence.*

Will be reformulated.

*Line 260- : I believe a correct representation of the climatological storm track is not enough to ensure spread and error to be proportional. For example, incorrect model response to the initialised/prescribed forcing (because of lacking model physics, e.g. parametrisation), or initialisation issues (errors, sampling, ecc) may also result in forecast errors or may affect the spread.*

We agree and will reformulate the sentence to better represent the point, that misrepresentation of the storm track will likely lead to anomalies in ensemble spread and hence misrepresentation of the model error.

*Line 280 : This last sentence is vague and inconclusive. Please specify how storm forecast and risk mitigation can benefit from the outcomes of this work.*

We intend to remove the sentence.

---

## Author Response (AR1)

**Referee #1**

We thank referee #1 for their review and various suggestions to improve the manuscript. In the following we will respond in to the different comments and explain the changes we made to the manuscript based on them. The reviewer's comments are in black italics, our responses are shown in blue. All line numbers and references in the referee's comments refer to the originally submitted manuscript, while line numbers in our responses refer to the revised version of the manuscript.

**GENERAL COMMENT**

*This manuscript analyses the relation between strong storms and forecast uncertainty, demonstrating how the prediction of storms, associated with strong geopotential height anomalies, affects the spread of ensemble forecasts in the Euro-Atlantic sector. Even though the ECMWF hindcast ensemble is small, as argued by the authors the results should naturally extend to larger ensembles.*

*This outcome is applied to the case of weak and strong stratospheric vortex state. The weaker frequency and depth of strong storms in the Euro-Atlantic sector (and Northern Europe) during weak stratospheric vortex conditions likely explains the decrease in uncertainty that follows these events.*

*Although the topic is original and interesting, the manuscript still requires substantial improvements in terms of linearity of the discussion and, ultimately, readability. Therefore, I encourage the authors to work specially on the structure of the text and on the precision of the langage, specially in the Introduction and Discussion sections. Other methodological and technical comments are included below. Once these issues have been addressed, I am willing to recommend publication of this work in Weather and Climate Dynamics.*

Thanks for your constructive review. We revised the text based on your comments and try to improve readability and precision, especially for the introduction and discussion.

**SPECIFIC COMMENTS**

*Introduction and Discussion :*
*From reading these sections, I was expecting the whole work to be focused on SPV impacts on uncertainty. The main results, however, are those relating the presence of strong storms and forecast uncertainty. Could you work on these in order to make the focus of your work clearer to the reader ?*

Thank you for this valuable feedback. We understand the need for a clearer focus throughout the manuscript. In response, we revised the Introduction and Discussion sections to better align with the main results of our study, which are centred around the general impact of strong storms on forecast uncertainty.

However, a significant area of applicability of our work lies in understanding how processes and teleconnections, such as the downward influence of the stratosphere, modulate forecast uncertainty through changes in the storm track. Originally, our study was strongly motivated by the anomalous spread signals following weak SPV states identified by Spaeth et al. (2024), but our results provide a more general quantification of the contribution of strong storms to forecast uncertainty.

*Both are rather dishomogeneous and difficult to follow. Connections between the different paragraphs and topics are often lacking.*

We revised the text in these sections to improve the flow and better guide the reader through our study.

*Figure 1 : for an equal comparison between the ensemble spread with and without storms the size of the ensemble must be consistent. For example, for the "with storms" case you can compute a line (or better, a distribution of lines) where the ensemble includes storm cases, plus a number of random members necessary to equal the size of the "without storm" ensemble.*

*Figure 5 : see comment to Figure 1.*

We agree with the referee that adequate statistical treatment of ensemble forecasts are crucial to obtain robust scientific results. However, the main requirement towards ensemble sizes is a sufficient sampling to be able to draw robust statistical conclusions. Given large enough samples, the ensemble sizes do not necessarily have to be equal to make a direct comparison of statistical characteristics. In our case the sample sizes are indeed small. However, by choosing the variance we are using an unbiased-estimate of the ensemble spread, which leads to a robust quantification irrespective of the ensemble size. Hence re-sampling the distribution by adding "random members" drawn from that distribution will (on average) not change the computed variance.

We tried to illustrate this in Figure R1, which shows plots similar to Fig. 1 and Fig. 5 in the manuscript. Figure R1a shows the evolution of ensemble spread for the case study hindcast discussed in Section 3. The black line shows the Z1000 ensemble spread over Northern Europe for the full ensemble (all 10 members). The blue line shows the spread for the same dataset, but "without storms", so all members disregarded that exhibit an active storm at a given leadtime. The spread shown by the blue line is hence computed for a dataset with usually fewer than 10 members. Fig. R1a also shows as green dashed line a version of this "spread without storms", where we artificially increased the ensemble size to 10 members again (by randomly re-drawing members from the "without storm" distribution). It can be seen that the blue and the green dashed lines are almost identical, except for very small deviations at certain leadtimes. Since Fig. 1 in the manuscript is mainly used to visualise the impact of storms on the spread for an exemplary case, we do not believe that these small deviations do not have a substantial impact on the conclusions that can be drawn from the figure or the associated confidence of the reader.

In addition, the deviations between the blue and the green dashed line are not systematic, i.e., they have both positive and negative sign. Averages over many cases will therefore not show any difference between the different ensemble sizes. This is illustrated in Fig. R1b, where we computed the climatological evolution of the ensemble spread as function of lead time (similar to the black line in Fig. 5 in the manuscript), but for re-sampled ensemble sizes (by randomly drawing a certain number of members from the initially 10-member ensemble). It can be seen that on average, the evolution of the spread is essentially independent of ensemble size, up to a small increase in variability (of the variance) for very small ensembles. As argued above, this further suggests that the reduced ensemble size by removing certain members from the dataset does not impact the estimation of ensemble spread. To keep the methodology as simple as possible, we therefore kept the approach used in the original study.

[Figure]

**Figure R1:** a) Same as Fig. 1 in the manuscript, but with an added line (green dashed) showing the ensemble spread for a dataset "without storm", but resampled to contain 10 ensemble members at each leadtime. The resampling is done by randomly drawing extra members from the set of the set of members "without storm" . b) Climatological evolution of ensemble spread (as shown in black in Fig. 5 of the manuscript) for different ensembles with reduced ensemble size (by randomly removing members from the dataset).

*Figures 6a and 7b : The fact that storm strength is high for small Pmin is confusing. Please address this issue as you think best.*

We changed the orientation of the y-axis in Fig. 7b, so it more reflects a 'storm strength', and added a corresponding note to the caption.

***TECHNICAL CORRECTIONS***

*Line 19 : "prescribed" or "initialised" in the case of S2S ?*

We modified the sentence accordingly.

*Lines 21-23 : Sentence is clear but quite convoluted. Please improve.*

We split the sentence.

*Lines 32-33 : Do you want to say that uncertainties in the position of the jet (and its associated baroclinic zone) lead to a large ensemble spread in the mid latitudes ? Is there a difference in laying the focus on*

*the jet or on the strong pressure gradients ? In my opinion, they just correspond to different aspects of the same weather feature.*

We agree with the referee that the jet and the strong meridional pressure gradient over the Atlantic are aspects of the same atmospheric feature. The reason we introduced the pressure gradient is, that it might be easier for the reader to understand how differences in the location of this gradient across different ensemble members can lead to an increase in ensemble spread. We revised the sentence to hopefully clarify the confusion.

*Line 34 : Maybe "synoptic-to-planetary scale" ?*

We adapted this formulation.

*Line 36 : Since you are introducing the focus of this work, maybe make new paragraph ?*

We added a line break.

*Line 57 : "NAO-like".*

Revised.

*Line 58 : "and OF THE North Atlantic storm track".*

Corrected.

*Line 59-61 : This sentence is rather vague. Please consider removing if outside of the scope and topic of this paper.*

We removed the sentence.

*Lines 90-93 : I find these two sentences confusing.*

We revised the paragraph to clarify the details of ECMWF real-time forecasts and hindcasts (L94-97).

*Why do you separate the 2020-21 hindcast from the hindcasts in previous years ?*

In our study we only utilise 10-member hindcasts, and none of the 50-member real-time forecasts. A practical reason is that we do not have storm track data available for the real-time runs. Another reason is, as explained in Section 2.1, that the impact of individual storms on the ensemble spread is easier to highlight in small ensembles. We revised the paragraph at the beginning of Section 2.1 to make sure it clarifies the distinction between real-time forecasts and hindcasts.

*Could you clarify the sentence "Each real-time initialisation provides hindcasts for the previous 20 years". Is this again 10-member hindcasts over the previous 20 years ?*

Yes, for each real-time forecast ECMWF runs 20 hindcasts initialised at the same day-of-year as the real-time run, but in one of the preceding 20 years. These hindcasts have 10 members (plus a control member, which we do not use). More information on the ECMWF subseasonal forecast system can be found here: https://confluence.ecmwf.int/display/S2S/A+brief+description+of+reforecasts

*It would be useful to state explicitly which terminology is used to distinguish the runs initialised in 2000-2019 and the ensemble runs from 2020-2021, if there is a distinction. Also, what does the term "forecasts" in the rest of the text refer to ? Note the inconsistency with the term "hindcasts" used to present the dataset.*

We clarified the matter by revising the introduction of "real-time forecasts" and "hindcasts", as they are typically used by ECMWF, and ensured these terms are used consistently within the manuscript.

*Line 123-124 : This sentence is an overstatement in my opinion. Maybe mention "intense" or "strong" storms ?*

Revised.

*Line 131 : Specify "Z1000 distribution".*

Revised.

*Line 135 : Specify the target domain from the beginning. Please add a motivation for choosing this region.*

Our target area is Northern Europe (55-67.5N and 0-20E), motivated by the region of reduced ensemble spread following weak stratospheric polar vortex events found by Spaeth et al. (2024a). We added this information to the text (L144 and 145).

Spaeth, J., Rupp, P., Garny, H. and Birner, T., 2024. Stratospheric impact on subseasonal forecast uncertainty in the Northern extratropics. *Communications Earth & Environment*

*Line 143 : How is the vicinity of the storm to the target domain defined ?*

We define the vicinity of the target domain as within 5 degrees around the border. We added this information (L152).

*Line 158 : The case study is for "Northern Europe"   right?*

Yes. We revised the statement.

*Line 161 : You have not yet defined the "Euro-Atlantic sector".*

We added this information (L175).

*Line 163 : Specify "a single storm CENTRE is therefore...". Further away from the centre, the percentage increase in spread is weaker.*

Revised.

*Line 172 : Could you justify why you go back to analysing Northern Europe, since the previous analyses in this subsection referred to the Euro-Atlantic sector ?*

As explained in the response for another comment, the main motivation for studying Northern Europe is the spread signal following weak stratospheric polar vortex states found by Spaeth et al. (2024), which was suggested to result from changes in storm activity. Figure 3 analyses the effect of strong storms on

the ensemble spread for storms detected within the entire Euro-Atlantic sector to show a more universal connection, which is not necessarily confined to Northern Europe.

*Lines 238-240 : A bit out of the blue here. Specify connection with previous or shift to a different paragraph.*

We reformulated the sentence to better embed it into the paragraph.

*Line 244-246 : Acknowledge briefly that reduction in uncertainty does not necessarily mean an improvement in the predictions.*

We added a brief statement regarding the improvement of predictions. In particular, we clarified that the spread-error connection holds under a perfect model assumption (L282-288). We further added a reference (Spaeth et al., 2024b) showing that the Z1000 ensemble spread correlates well with the forecast error over northern Europe in the ECMWF model (L243-245).

Spaeth, J., Rupp, P., Osman, M., Grams, C.M. and Birner, T., 2024. Flow-dependence of ensemble spread of subseasonal forecasts explored via North Atlantic-European weather regimes. *Geophysical Research Letters*

*Line 257-258 : It's a bit strange to consider Rossby waves and storms in opposition, since storms can be seen as Rossby-wave troughs. Please consider this while revising the text.*

We agree that synoptic Rossby waves and storms are highly correlated and, in particular, storms often develop in the troughs of Rossby waves during their non-linear breaking phase. In this study, we distinguish conceptually between the quasi-linear propagation stage of Rossby waves and storms since the two flow features affect forecast uncertainty in slightly different ways. Storms are usually isolated features, which at subseasonal leadtimes can occur in one ensemble, while another one does not show any signs of storms. We ensured the revised Discussion sections clarifies this matter (L277-281).

*Line 259 : Correct the syntax of the last part of the sentence.*

We reformulated the sentence.

*Line 260- : I believe a correct representation of the climatological storm track is not enough to ensure spread and error to be proportional. For example, incorrect model response to the initialised/prescribed forcing (because of lacking model physics, e.g. parametrisation), or initialisation issues (errors, sampling, ecc) may also result in forecast errors or may affect the spread.*

We agree and made the revised discussion better represente the point, that misrepresentation of the storm track will likely lead to anomalies in ensemble spread and hence misrepresentation of the model error (L282-288).

*Line 280 : This last sentence is vague and inconclusive. Please specify how storm forecast and risk mitigation can benefit from the outcomes of this work.*

We removed the sentence.

**Referee #2**

We thank referee #2 for their review, questions and suggestions to improve the manuscript. In the following we will respond in to the different comments and explain the changes we made to the manuscript based on them. The reviewer's comments are in black italics, our responses are shown in blue. All line numbers and references in the referee's comments refer to the originally submitted manuscript, while line numbers in our responses refer to the revised version of the manuscript.

*This paper looks at the impact that storms have on ensemble spread of S2S forecasts, focussing on Northern Europe/the North Atlantic region. The results show that the occurrence of strong storms contributes around 20% of the Z1000 spread. The study then moves on to look at the impacts of strong and weak SPV states on storms and ensemble spread. Overall the paper is well written, with clear and useful figures, and a logical structure. I have only a couple of minor comments.*

Thanks for highlighting the strengths of our paper and the positive feedback.

*Minor comments:*

*Fig 1: it is interesting to see that on some days the spread without storms is (albeit slightly) higher than the spread with storms: do you have any ideas why this would happen? I can see that it happens when the Z1000 spread of the active storm members is within the range of other members, but in a dynamical sense how would you interpret this?*

This is indeed an interesting question. Situations where the presence of a storm does not lead to a significant increase in spread, or even a decrease, is indeed, as you explained, if the Z1000 index of the corresponding ensemble member is close to the ensemble mean. One practical scenario that could lead to such a situation if the centre of a relatively weak storm is detected inside the detection area, while most of the Z1000 anomaly of that storm lies outside of the area. Such a situation is for example happening at leadtime day 15 in the case study discussed in Section 3 of the manuscript (hindcast initialised 31 December 2007). In this case, we identify a storm within the detection area for only for 1 of 10 members (see Fig. 1 in the manuscript). Removing this member associated with a storm from the dataset very slightly increases the ensemble spread.
The synoptic situation of selected members for this case are shown in Figure R2.1. Member 4 of this hindcast shows a storm at the very edge of the detection area (green box), but of the Z1000 anomaly lies outside of the area used to compute the Z1000 spread (purple box). In addition, members 1 and 2 show storms with centres just outside of the green box, but with Z1000 anomalies reaching inside the purple box, hence these members will likely show more negative Z1000 extremes than member 4, while not being associated with a storm. However, these situations are rather rare. The motivation for choosing a slightly larger detection area than area used for Z1000 spread analysis is based on the synoptic size of storms and to reduce the impact of situations like shown in Fig. R2.1b, where a storm centre lies outside of the target area, but the storm extends inside.

We added a short note to the discussion of Fig. 1 to address this issue (L155-159).

[Figure]

**Figure R2.1:** Z1000 anomalies (contours) and associated spread anomalies (shading) for selected member at lead-time day 15 of the hindcast with initial conditions from 31st December 2007. Markers indicate the centre location of strong storms in the respective member. Marker sizes scaled with storm strength. Purple thick box indicates the Northern European region used for averages in the manuscript, green thin box shows the region used to identify members with active storm.

*Fig 4 a and b: I didn't entirely understand what is plotted here or follow the interpretation about this - Please could you try to clarify more in the text.*

Figure 4 shows a composite over 'spread peaks', which are identified as time steps in a hindcast where the ensemble spread anomaly is extremely high. The red curve in Fig. 4a shows the composite mean of the spread anomaly indeed exceeding the 95[th] percentile for these events. The panel further shows the distribution of Z1000 anomaly over Northern Europe around these events. While the Z1000 distribution is symmetric around 0 long before or after the events, it is highly skewed during the event (lag=0). The reason is that strong storms (which are a main driver of these extreme spread events) are associated with extremely negative Z1000 anomaly.

To further substantiate the hypothesis that strong storms drive such spread events, Figure 4b shows the storm density and strength in the area. At the time of the event (lag=0), both metrics are significantly increased, suggesting that both physical characteristics contribute to the extreme spread (more storms and stronger storms).

We revised the discussion of this figure to clarify our interpretation of the associated results (L186-189).

***Typos:***

*Fig 4 caption: anomay – should be anomaly*
Corrected.

*Fig 5 caption: Evolution of …*
Corrected.